# Synthesis, Structure and Photochemistry of Dibenzylidenecyclobutanones

**DOI:** 10.3390/molecules27217602

**Published:** 2022-11-05

**Authors:** Marina V. Fomina, Alexandra Y. Freidzon, Lyudmila G. Kuz’mina, Anna A. Moiseeva, Roman O. Starostin, Nikolai A. Kurchavov, Vyacheslav N. Nuriev, Sergey P. Gromov

**Affiliations:** 1Photochemistry Center of RAS, FSRC “Crystallography and Photonics”, Russian Academy of Sciences, Novatorov Str. 7A-1, 119421 Moscow, Russia; 2N.S. Kurnakov Institute of General and Inorganic Chemistry, Russian Academy of Sciences, Leninskiy Prosp. 31, 119991 Moscow, Russia; 3Department of Chemistry, M.V. Lomonosov Moscow State University, 119991 Moscow, Russia

**Keywords:** dienones, synthesis, X-ray diffraction analysis, NMR spectroscopy, cyclic voltammetry, electronic spectroscopy, quantum chemical calculations

## Abstract

A series of symmetrical dibenzylidene derivatives of cyclobutanone were synthesized with the goal of studying the physicochemical properties of cross-conjugated dienones (ketocyanine dyes). The structures of the products were established and studied by X-ray diffraction and by NMR and electronic spectroscopy. All the products had *E*,*E*-geometry. The oxidation and reduction potentials of the dienones were determined by cyclic voltammetry. The potentials were shown to depend on the nature, position, and number of substituents in the benzene rings. A linear correlation was found between the difference of the electrochemical oxidation and reduction potentials and the energy of the long-wavelength absorption maximum. This correlation can be employed to analyze the properties of other compounds of this type. Quantum chemistry was used to explain the observed regularities in the electrochemistry, absorption, and fluorescence of the dyes. The results are in good agreement with the experimental redox potentials and spectroscopy data.

## 1. Introduction

The enormous synthetic potential of the carbon–carbon bond conjugated to a carbonyl group has been long and successfully used in organic chemistry. A typical example is the Michael reaction and its numerous varieties [1]. The introduction of two double bonds in conjugation with a keto group enables some additional reactions [2]. These compounds, called cross-conjugated dienones, ketocyanine dyes, or diarylidene ketone derivatives, attract researchers’ attention for their versatile synthetic chemistry and extensive applicability, first of all, in biology and medicine [3,4,5]. One more potential application of cross-conjugated dienones is the design of photoactive materials [6,7,8,9].

The known methods for the synthesis of these compounds have been developed primarily for dienones derived from cyclopentanone or cyclohexanone and various aromatic aldehydes. Their homologues with larger or smaller rings have been rarely addressed, which may be attributable to the poor availability of the corresponding cycloalkanones as well as side reactions decreasing the yield of the target compound. The first attempts to synthesize a diarylidene derivative of cyclobutanone through aldol condensation, catalyzed by 60% KOH in ethanol, were described in [10]. However, it was shown later that the dienone obtained in that paper was a dimer of 2,4-dibenzylidenecyclobutanone. Upon the action of the hydroxide, a fast dimerization of the resulting dienone takes place. It was found that the susceptibility to dimerization increases with the temperature and the basicity of the reaction mixture [11,12]. Another possible reason for the low reaction yield can be side processes arising from the high reactivity of cyclobutanone, which is susceptible to self-condensation in basic media [13,14]. Therefore, only single examples of 2,4-dibenzylidenecyclobutanones have been characterized in the literature [11,12,13,14,15]. Meanwhile, recently, cross-conjugated dienones derived from cyclobutanone have attracted attention as sensitizers to the generation of singlet oxygen for the photodynamic therapy of cancer [16,17].

Apart from participation in additional reactions, double bonds are also responsible for two important properties of dienones. First, a prominent feature of dienones is the existence of *E*- and *Z*-isomers, which can be interconverted under the action of various stimuli such as light, acids, or transition metals [18,19,20,21,22]. Most often, the *E*,*E*-isomer is most stable in the series of cyclopentanone and cyclohexanone derivatives. The proportion of *E*,*Z*- and *Z*,*Z*-isomers increases with the increasing ring size [3,20]. For cyclobutanone-derived dienones, information of this type is scarce.

One more feature of this class of compounds is the ability to undergo [2 + 2]-photocycloaddition (PCA) reaction [20,23,24]. In the case of free dienones, this reaction can proceed both in the crystal and in solution. The possibility and the stereoselectivity of PCA can be controlled by the supramolecular preorganization of double bonds, which may ensure the most appropriate geometry for the preceding dimer pair [25,26,27]. In the case of crystalline pyridine derivatives, such reactions can be accomplished using supramolecular templating (initiation) by metal complexes [28], resorcinol (pyridine-containing monoenones and acyclic dienones [29]), or silver ions (only pyridine-containing acyclic dienones were studied) [30,31] The possibility of PCA for 2,4-dibenzylidenecyclobutanones has not been discussed in the literature.

With the purpose of developing photoswitchable supramolecular systems, we started a comprehensive study of cross-conjugated dienones containing crown-ether moieties as ionophore substituents [32,33]. This research is meant to design hybrid molecules combining two functional moieties: an ionophore able to bind metal and ammonium cations and guest molecules, and a photoswitchable moiety needed for controlled binding, using light as an energy source.

Gaining a more in-depth understanding of the involved photochemical transformations required a detailed investigation of the properties and physicochemical characteristics of the model compounds. In addition, it was necessary to elucidate the possibility of a PCA reaction in the crystal without supramolecular preorganization of the reacting double bonds.

Here, we synthesized and studied dienones **1a**–**f**, differing in the nature and number of alkoxy, alkylthio, and dialkylamino substituents (Figure 1). The structures in question are chromophore models of crown ethers and aza/thia-crown ethers. Cyclobutanone was chosen as the central moiety as an analog of cyclopentanone derivatives, such as **2,** and cyclohexanone derivatives, addressed in our previous studies [34,35].

The structures of compounds were determined by X-ray diffraction analysis and by NMR and electronic spectroscopy. X-ray diffraction analysis was used to find out whether the arrangement of double bonds of neighboring molecules is favorable for the PCA to occur in the crystal without a supramolecular effect. Cyclic voltammetry was employed to determine the oxidation and reduction potentials in order to elucidate the dependences of the energy characteristics of the molecules on the position, nature, and number of substituents in the benzene rings. Quantum chemistry was used to find the preferred conformation of **1c**, to relate the electrochemical and photochemical data, and to elucidate the mechanism of luminescence emission and quenching in **1a**–**f**. In addition, we attempted to identify a correlation between the photophysical and electrochemical characteristics of dienones **1a**–**f**.

## 2. Results and Discussion

### 2.1. Synthesis of **1a**–**f**

Compounds **1a**–**f** and **2** were synthesized by alkaline aldol-crotonic condensation of cyclobutanone and cyclopentanone, respectively, with two equivalents of substituted benzaldehyde (Claisen–Schmidt reaction, Figure 1), carried out similarly to the reported procedures [10,11,12,13,14,15,20,34].

Compounds **1a**–**f** were isolated as bright-colored crystalline solids. X-ray diffraction data were obtained for all the compounds, except for **1a** (for details, see below), and indicated that all the dienones were formed as *E*,*E*-isomers.

A similar conclusion can be drawn from the data of NMR spectroscopy. Indeed, the chemical shifts of the olefinic protons of dienones **1a**–**f** in the 7.01–7.19 ppm range attest to the *E*,*E*-isomers [20].

### 2.2. X-ray Diffraction Analysis

The crystals of compounds **1b**–**f** and **2** (*syn*,*syn*- and *anti*,*anti*-conformers) suitable for X-ray diffraction study were grown from MeCN solutions. Each of these crystals was subjected to X-ray diffraction analysis. The solved structures of **1b**-**f** supported the results of the NMR spectroscopic study, indicating that only *E*,*E*-isomers formed for the dibenzylidenecyclobutanone molecules. The structures of compounds **1b**–**f** are depicted in Figure 2.

In the crystals of **1b**–**d,** the molecule occupies a special position on a twofold axis, whereas in **1e**, the molecule is in a general position. In the crystal of **1f**, there are two crystallographically independent molecules [**1f** (A) and **1f** (B)]. In all of the compounds, the molecular skeleton is nearly planar.

Selected bond lengths and bond angles for **1b**–**f** are listed in Appendix A.

The geometric parameters of both the independent molecules of **1f** are actually identical. Figure 3 shows a superposition of these molecules. Only small differences in the torsion angles associated with the benzyl-ring orientation can be observed.

It is seen from the data of Appendix A that the most important corresponding geometric parameters are well reproduced for molecules **1b**–**f**, containing the central four-membered ring. In particular, a significant deformation of the exocyclic angles at the C2 atoms of the four-membered ring is observed, with the C1-C2-C4 angle being ~10° smaller than the C3-C2-C4 angle. Some deformation of the cyclobutane ring angles is also noted: the C2-C3-C2 angle is smaller than the other angles of the ring. One would assume that the above-mentioned non-equivalence in the exocyclic bond angles might be minimized via the rotation of the benzene rings around the C4-C5 bonds. However, this is not observed. The reason may be a significant conjugation over the whole Ph-C=C-C(O)-C=C-Ph moiety, although the bond length distribution does not display an elongation of the carbonyl bond and C=C double bonds. These bond lengths are close to the normal values for localized bonds. The 1-2-4-5 moieties are nearly planar, the corresponding torsion angles are close to 180° or 0°.

It is pertinent to compare the data of Appendix A with the corresponding data for compounds **2** based on a five-membered central ring (2,5-dibenzylidenecyclopentanones) (*syn*,*syn*- and *anti*,*anti*-conformers). Figure 4 shows the molecular structures of two such compounds (*syn*,*syn*-**2** and *anti*,*anti***-2**), while Appendix A lists selected geometric parameters of the molecules.

The geometric parameters from Appendix A demonstrate close similarity for both (*syn*,*syn*- and *anti*,*anti*-) conformers of **2**. These data well agree with the corresponding values for other derivatives of dibenzylidenecyclopentanones [34,36]. Moreover, the most important features are reproduced for both five- and four-membered ring systems. In particular, this is the planarity of the molecule, bond-length alternation within the Ph-C=C-C(O)-C=C-Ph moiety, and also the same type of exocyclic angle deformation.

Earlier, it was established that the vast majority of conjugated planar molecules form six canonic types of crystal packings, two of which (stacking and parallel-dimeric packing) are favorable for the solid-state [2 + 2] photochemical reaction [27]. In spite of the planarity of these molecular systems (**1b**–**f** and **2**), they do not form the canonic packing motifs in the crystalline state. Relatively close contacts between the ethylene bonds of the adjacent molecules were observed only in the structures of **1b**–**d** (Figure 5).

The distance between the ethylenic bonds in **1c** is longer than 5 Å. In all the cases, the molecules in a pair are shifted with respect to one another in parallel planes. The distances between the ethylenes are much longer than 4.2 Å, which makes the PCA reaction in these molecules impossible [27]. Apparently, the PCA reaction of compounds **1b**–**f** and **2** in the solid state requires the use of a supramolecular template.

The results of the quantum chemical calculations are in line with the X-ray diffraction data. The 5-4-3-2 torsion angles decrease from 25° for the unsubstituted dienone **1a** to 16–17° for **1e**,**f**. This trend is correlated with the variation of the electron-donating properties of the substituents in the benzene rings of the dienones.

### 2.3. NMR Spectroscopy

NMR spectroscopy can serve to elucidate the fine structure of organic molecules and molecular assemblies in solutions [37]. The NMR data are rarely compared with the structures of compounds known from a crystallographic analysis because it is impossible to obtain high-quality crystals for the whole series. In the case of dienones, the determination of their conformational behavior in solutions is especially important for the prediction and determination of the structures of supramolecular systems based on bis-crown-containing dienones [32]. Therefore, we studied the structural characteristics of (*E*,*E*)-dienones **1c** and **2** using various NMR techniques.

In the crystalline state, (*E*,*E*)-tetramethoxydienone **1c** and (*E*,*E*)-tetramethoxydienone **2** occur as nearly planar symmetrical *syn*,*syn*-, *syn*,*syn*-, and *anti*,*anti*-conformers, respectively, which is probably attributable to the requirement of the close packing of molecules. Upon dissolution, a fast, conformational equilibrium may be established between the symmetrical *syn*,*syn-* and *anti*,*anti*- and unsymmetrical *syn*,*anti*-conformers (Figure 4), by analogy with the previously studied bis-crown-containing stilbenes and distyrylbenzenes [38,39].

The NOESY spectrum of compound **1c**, which is given in Appendix A (the atom numbering differing from IUPAC rules is presented in Figure 1), and the spectrum of **2** in CD_2_Cl_2_ show averaged signals from the different conformers. The spectrum of dienone **1c** has an intense cross-peak, corresponding to the through-space intramolecular interaction of the H(2′) protons of the benzene ring with the H(3) methylene protons of the cyclobutanone moiety, and a less intense cross-peak between these protons and the H(6′) aromatic protons. It is noteworthy that the spectrum contains no cross-peaks between the H(2′) and H(6′) protons of the benzene ring and the H(α) protons of the ethylene bonds (Appendix A). This overall spectral pattern can be interpreted by assuming that the *syn*,(*syn*/*anti*)-conformers predominate in the equilibrium.

The NOESY spectrum of dienone **2** also exhibits intense intramolecular cross-peaks, indicating the coupling of the H(2′) and H(6′) protons of the benzene ring and the H(3) methylene protons of the cyclopentanone moiety. Apart from the indicated cross-peaks, the H(6′) aromatic protons of dienone **2** give a small NOESY cross-peak with the H(α) ethylene protons [35]. 

Although the NOESY spectra do not enable an accurate estimate of the contribution of each conformer of **1c** and **2** to the equilibria due to the strong coupling of the methylene protons with both type H(2′) and type H(6′) protons in all the conformers, the revealed spectral features provide the conclusion that *syn*,(*syn*/*anti*)-conformers predominate in the solution in both cases. 

Using the data on the energies of stable structures for these compounds found by the FireFly program package, we calculated the theoretical ratio between the three possible conformers for dienones **1c** and **2** (Figure 6).

The theoretical ^1^H NMR spectra of conformers of (*E*,*E*)-dienone **1c** and **2** simulated using their mole fractions are in good agreement with the experimental data (Table 1); in particular, the calculations predict higher field positions for the H(5′) *meta*-protons than for the H(6′) and H(2′) *ortho*-protons. The calculated distances between the H(2’), H(6’) protons and the H(α) methine protons, or between the H(2’), H(6’) protons and the H(3) methylene protons, are sufficient for the manifestation of NOE interactions, see Figure 7. The conformer models predict the possible appearance of NOESY cross-peaks between these groups of protons. 

The predominance of *syn*,(*syn*/*anti*)-conformers of dienone **1c** can be explained by competition between the steric factor and the stabilization via dipole–dipole interactions with polar molecules of the medium. According to calculations, these conformers have the highest theoretical dipole moments. Despite the sterically unfavorable conformation, additional stabilization is brought about by strong interactions with the molecules of the medium. Ongoing to the *anti*,*anti*-conformation, the energy benefit caused by the decrease in the steric strain is counterbalanced by the trend towards a decrease in the dipole moment of the molecule, which leads to destabilization in polar dichloromethane. In the case of *anti*,*anti*-conformer, the dipole moment decreases by a few units. As a result, destabilization starts to predominate. 

### 2.4. Electrochemistry

A cyclic voltammetry (CV) study of compounds **1a**–**d**,**f** was carried out on a cleaned surface of a glassy carbon (GC) electrode in MeCN in order to reveal the effect of substituents in the aromatic rings of cross-conjugated dibenzylidene cyclobutanones on the frontier orbital energies in comparison with the same characteristics of the cyclopentanone- and cyclohexanone-based dienones studied previously [34,35]. The CV curves were recorded starting from 0 V and moving towards the cathodic and anodic potentials. Table 2 gives the first peak potentials determined in the MeCN for comparison of the electrochemical characteristics with the data of other physicochemical investigations in the same solvent. The same Table 2 presents the differences (shifts) of the reduction and oxidation potentials of the substrates relative to those of the unsubstituted compound **1a**.

The cathodic and anodic processes of compounds **1a**–**d**,**f** are irreversible (Figure 8a). The compounds **1b**–**d**,**f** containing substituents in the aromatic rings are reduced with more difficulty than the unsubstituted **1a**. The first cathodic peak potentials of dienones **1b**–**d**,**f** are shifted by 140–310 mV to more negative values relative to that of **1a** (Table 2, Figure 8a). The most pronounced electron-donating effect on the cathodic potential is observed for the diethylamino group (310 mV), while the least pronounced effect was found for the methylthio group (50 mV).

The cathodic peak potentials of dibenzylidene cyclobutanones shift to the anodic region relative to the potentials of similar cyclopentanone derivatives and, to a larger extent, relative to cyclohexanone derivatives (Table 3 [34,35]). This may be due to the more planar molecular geometry of the dibenzylidene cyclobutanone derivatives, resulting in a higher degree of conjugation in comparison with similar cyclopentanone and cyclohexanone derivatives.

The electron-donating ability of the *para*-substituents in the benzylidene moieties of dienones has a more pronounced effect on the anodic potentials than on the cathodic potentials. The shifts of the oxidation peak potentials of **1b**–**d**,**f** to lower anodic values relative to that of the unsubstituted dienone **1a** are 430–1190 mV.

The frontier orbital energies corresponding to the ionization potentials (HOMO) and electron affinities (LUMO) calculated by quantum chemical techniques are summarized in Table 2. The theoretical values are correlated with the oxidation and reduction potentials and qualitatively reproduce the pattern of variation of these values in the series of dienones found experimentally (the shift relative to **1a**). The calculation confirms the conclusion about the effect of the electron-donating properties of the substituents on the shifts of the cathodic and anodic peaks and the band gap ΔE. The ionization mechanism is similar to that found for dienones based on cyclopentanone and cyclohexanone [34,35]. An electron is removed from the HOMO-1 quasi-degenerate with HOMO and is accepted by the LUMO. 

### 2.5. Photophysics

Electronic absorption and fluorescence spectra were measured to determine the effect of substituents on the spectral properties of dienones **1a**–**f** (Table 4, Figure 9, Appendix A).

All dienones exhibit a long-wavelength absorption band (LWAB) (λ_max_ from 341 nm for **1a** to 481 nm for **1f**) and several more bands in the shorter wavelength range. LWABs can be assigned to the HOMO–LUMO transitions [40,41], while the shorter wavelength bands can be attributed to local electron transitions in the aromatic rings. Attention is attracted by the qualitative dependence of the LWAB maxima on the electron-donating ability of the substituents in the *para*-position of the benzylidene moieties. The LWABs of *para*-dialkylamino-substituted **1e**,**f** are red-shifted relative to that of the unsubstituted **1a,** to the greatest extent, which is in good agreement with the high electron-donating ability of the substituents and is correlated with the lowest oxidation potentials among the considered series (Table 4). In the case of *meta*-methoxy-substituted compound **1c**, the LWAB maximum shifts to 401 nm, and the oxidation potential increases relative to those of **1e**,**f**. The further red shift of LWAB of dienone **1d** is also accompanied by increasing *E*_ox_ and is caused by the higher electron-donating ability of the SMe substituent.

The calculated absorption and fluorescence spectra are in qualitative agreement with the experiment and reproduce the trend in the series. The first electron transition for all the dyes, except for **1a**, is the π-π* transition. In **1a**, the first electron transition is the dark *n*-π* transition. This accounts for the absence of fluorescence in dye **1a**. The corresponding orbitals are depicted in Figure 10. Generally, the red shifts of the LWABs of dienones **1a**–**f,** relative to those of dienones based on cyclopentanone and cyclohexanone, are caused by the more planar geometry of the π-electron system of cyclobutanones.

The dependence of fluorescence properties of compounds **1a**–**f** on their oxidation potentials is generally similar to the λ_max_(LWAB)–*E*_ox_ dependence. The brightest and longest-wavelength fluorescence is observed for the amino derivatives **1e**,**f** (Table 4).

The trends of variation of the fluorescence properties of cyclobutanones **1a**–**f** differ from those described for a series of related compounds [35]. Unlike cyclopentanone- and cyclohexanone-based dienones, most compounds with electron-donating substituents **1b**–**f**, except for **1a,b**, exhibit fluorescence.

According to the calculations, the *E*,*E*-isomer is dominant for each of the compounds in the S0 state. In the S1 state, a relatively fast *E*-*Z* isomerization can proceed, resulting in several isomers being present in comparable amounts (Table 5). Dienone **1d** contains significant amounts of all three possible isomers. However, unlike the other series, the dominant isomers of compounds **1b**–**f** are characterized by short radiation lifetimes. Therefore, fluorescence is the preferable deactivation pathway for the excited states as compared to the nonradiative deactivation pathways such as internal conversion or intersystem crossing (Table 4). This is in line with the experimental data.

We have analyzed the factors affecting the fluorescence quantum yield in the cyclobutanone series. Our calculations showed that the dyes in study have two low-lying excited states of the π-π* and ***n***-π* types. The ***n***-π* transition is forbidden, and the corresponding excited state is dark, while the π-π* transition corresponds to an intense absorption and emission band. In the unsubstituted dienone 1a, the dark ***n***-π* state lies below the bright π-π* state, which results in the lack of emission. Previously, we observed a similar picture in a cyclohexanone analog of dienone 1a and attributed its lack of fluorescence to the same reason [35,42].

The main mechanism of deactivation in the studied dienones is the structural relaxation resulting in the distortion of the conjugation. Our calculation showed that the twisting of the formally double bond during *E*-*E-E*-*Z* isomerization is the most probable way (Figure 11a). The (*E*,*E*)-dienone is excited, and its relaxation on the S1 potential energy surface leads to the twisting of the formally double bond. After overcoming a relatively low barrier, the molecule reaches the S1-S0 conical intersection, which corresponds to the ~90-degree twist. This conical intersection is of the funnel type, and further relaxation may proceed on the S0 potential energy surface towards either of the (*E*,*Z*) or (*E*,*E*) forms. We have built the potential energy profiles for the ground and two lowest excited states (Figure 11b shows the general scheme common for all the compounds in the series). Note that TDDFT poorly describes the behavior of the potential energy surfaces near conical intersections, therefore, these profiles can only be used for qualitative conclustions.

All the profiles show a conical intersection (CI) between the S0 and S1 states and two barriers on the S1 state, from the *E*,*E* form to CI and from the *E*,*Z* form to CI. The energy of the CI point is lower than the energy of either the *E*,*E* or *E*,*Z* minima on the S1 surface. This means that the relaxation should proceed nonradiatively, hindered only by a barrier. The depth of the CI relative to the deepest minimum (*E*,*E* form) is the driving force of the relaxation. This depth decreases from **1a** to **1f** (Table 6). From the CI point, the molecule nonradiatively goes to the ground state. The transition states (TSs) separate the S1 minima from the CI point. A TDDFT calculation gives the TS position at φ ~ 60–70°. The barriers hinder the rotation, and their height increases from **1a** to **1f** (Table 6). This means that increasing the donor capacity of the substituent makes nonradiative relaxation less probable, both thermodynamically and kinetically, and facilitates fluorescence.

The calculated activation energies were used to assess the isomerization rate constants (Table 7). A comparison of the characteristic isomerization times with the radiative lifetime shows that *E*-*E-E*-*Z* isomerization pathway can explain partial fluorescence quenching in **1b** and the lack of fluorescence in **1a**. The left barrier on the S1 potential energy surface can be overcome via thermal vibrations, and rotation around the double bond can be activated. Increasing the donor capacity of the substituents from **1a** to **1f** hinders isomerization, with almost unchanged τ_r_, which leads to the noticeable fluorescence of **1c**–**f**, in excellent agreement with the experiment.

In addition, we considered an alternative deactivation channel proposed for styryl dyes in [43], namely, the twisting of the phenyl ring around the formally single bond. Our calculations showed that, in dienones, such twisting gives neither stable structures on the S1 surface nor S1-S0 conical intersections.

The absorption and emission of compound **1f** and its aza-crown ether analog are described in detail elsewhere [42].

### 2.6. Correlations

Previously, we studied the relationships between a number of calculated and experimental characteristics for a broad range of compounds, including dienones. In particular, we demonstrated the existence of a linear relationship between the electrochemical and spectrophotometric characteristics of dienone molecules [34,35,44,45].

In order to establish the relationship between the data of absorption spectra and electrochemical measurements, and to elucidate the possible dependences of these results on the electronic properties of the substituents in the benzene rings of dienones **1a**–**f**, we carried out a correlation analysis of the relationship between the long-wavelength absorption maximum and oxidation/reduction potential difference (Figure 12).

The results presented here indicate that the energy characteristics of frontier orbitals in a series of related compounds, such as cross-conjugated dienones, can be adequately described by both electrochemical and spectrophotometric results, despite the irreversibility of the electrochemical reduction step. The results obtained in this work can be used to analyze new dienones and to design the desired characteristics of new molecules. 

## 3. Materials and Methods

### 3.1. Materials

MeCN (extra high purity, water content < 0.3%, Cryochrom) was used to prepare the solutions. The Bu_4_NClO_4_ (≥99%, for electrochemical analysis) was purchased from Sigma-Aldrich and used as the supporting electrolyte. The cyclobutanone, cyclopentanone, benzaldehyde, 4-methoxybenzaldehyde, 3,4-dimethoxybenzaldehyde, 4-(methylthio)benzaldehyde, 4-dimethylaminobenzaldehyde, and 4-diethylaminobenzaldehyde (Sigma–Aldrich) were used as received. The EtOH (chemically pure grade) was used without additional purification, (2*E*,5*E*)-2,5-bis(3,4-dimethoxybenzylidene)cyclopentanone (**2**) was prepared as described previously [34]. The DC-Alufolien Aluminiumoxid 60 F254 neutral was purchased from Merck. 

#### Synthesis of 2,6-Dibenzylidenecyclobutanone Derivatives 1a-f

**(2*E*,4*E*)-2,4-Dibenzylidenecyclobutanone (1a).** A mixture of cyclobutanone (70 mg, 1 mmol) and benzaldehyde (212 mg, 2 mmol) in 75% EtOH (875 μL) was added dropwise with stirring at 5 °C to a 0.03 M solution of NaOH in 75% EtOH (3.5 mL). The reaction mixture was stirred for 35 min, and a dilute solution of acetic acid in EtOH was added to pH 6. The precipitate thus formed was collected on a filter, washed with an EtOH–H_2_O mixture (1:1 *v*/*v*), and dried in air. This gave 30.1 mg of dienone **1a** as a light-yellow crystalline powder. Yield 12%, m.p. 185–187 °C (cf. Ref. [11]: 191–192 °C). ^1^H NMR (δ, ppm, *J*/Hz): 3.89 (m, 2 H, C(3)H_2_), 7.19 (m, 2 H, 2 C(α)H), 7.39–7.46 (m, 6 H, 2 H(3′), 2 H(4′), 2 H(5′)), 7.59 (dd, 4 H, 2 H(2′), 2 H(6′), *J* = 1.3, *J* = 6.7). ^13^C NMR (δ, ppm): 36.57 (C(3)), 127.35 (2 C(α)H), 129.53 (2 C(3′), 2 C(5′)), 130.43 (2 C(2′), 2 C(6′)), 130.58 (2 C(4′)), 135.37 (2 C(1′)), 146.95 (C(2), C(4)), 190.93 (C(1)). UV–vis (MeCN) *λ*_max_ 341 nm (*ε* = 38,000 M^−1^ cm^−1^), 356 nm (*ε* = 33,600 M^−1^ cm^−1^). Fluorescence: none. IR (CH_2_Cl_2_, ν): 1714 cm^−1^ (C=O). HRMS (ESI+) *m/z* calcd for C_18_H_15_O [M + H]^+^: 247.1117. Found: 247.1125. Elemental analysis calcd (%) for C_18_H_14_O 0.25H_2_O: C, 86.20, H, 5.83. Found: C, 86.06, H, 5.88.

**(2*E*,4*E*)-2,4-Bis(4-methoxybenzylidene)cyclobutanone (1b).** A 3 *M* solution of NaOH (250 μL) in an EtOH–H_2_O mixture (2:1 *v*/*v*) was added with cooling and stirring to a mixture of cyclobutanone (0.35 mg, 0.5 mmol) and 4-methoxybenzaldehyde (0.143 mg, 1 mmol) in EtOH (100 μL). The reaction mixture was kept at 5 °C for 5 h, and the precipitate thus formed was collected on a filter, washed on the filter with an EtOH–H_2_O mixture (2:1 *v*/*v*), and recrystallized from EtOH containing some CH_2_Cl_2_. This gave 89 mg of dienone **1b** as a yellow crystalline powder. Yield 58%, m.p. 190–193 °C (from EtOH–CH_2_Cl_2_) (cf. Ref. [11]: 193–194 °C). ^1^H NMR (δ, ppm, *J*/Hz): 3.78 (br.s, 2 H, C(3)H_2_), 3.83 (s, 6 H, 2 Me), 6.95 (d, 4 H, 2 H(3′), 2 H(5′), *J* = 8.5), 7.11 (m, 2 H, 2 C(α)H), 7.53 (d, 4 H, 2 H(2′), 2 H(6′), *J* = 8.5). ^13^C NMR (δ, ppm): 36.00 (C(3)), 55.97 (2 C, 2 Me), 115.02 (2 C(3′), 2 C(5′)), 126.63 (2 C(α)H), 128.16 (2 C (4′)), 132.09 (2 C (2′), 2 C (6′)), 144.69 (2 C(1′)), 161.79 (C(2), C(4)), 190.73 (C(1)). UV–vis (MeCN) *λ*_max_ 385 nm (*ε* = 40,400 M^−1^ cm^−1^). Fluorescence: none. IR (CH_2_Cl_2_, ν): 1714 cm^−1^ (C=O). HRMS (ESI+) *m/z* calcd for C_20_H_19_O_3_ [M + H]^+^: 307.1328. Found: 307.1337. Elemental analysis calcd (%) for C_20_H_18_O_3_: C, 78.41, H, 5.92. Found: C, 78.22, H, 5.94.

**(2*E*,4*E*)-2,4-Bis(3,4-dimethoxybenzylidene)cyclobutanone (1c).** A mixture of cyclobutanone (35 mg, 0.5 mmol) and 3,4-dimethoxybenzaldehyde (167 mg, 1 mmol) in EtOH (650 μL) was added dropwise with stirring to a 0.15 M solution of NaOH in 75% EtOH (2.75 mL). The reaction mixture was stirred at room temperature for 4 h, and the precipitate was collected on a filter, washed on the filter with water, and dried in the air. This gave 80.5 mg of dienone **1c** as a yellow-orange crystalline powder. Yield 46%, m.p. 187–190 °C (cf. Ref. [12]: 191–193 °C). ^1^H NMR (δ, ppm, *J*/Hz): 3.81 (m, 2 H, C(3)H_2_), 3.86 and 3.87 (2 s, 12 H, 2 3′-MeO, 2 4′-MeO), 6.92 (d, 2 H, 2 H(5′), *J* = 8.2), 7.06 (d, 2 H, 2 H(2′), *J* = 1.8), 7.10 (m, 2 H, 2 C(α)H), 7.19 (dd, 2 H, 2 H(6′), *J* = 8.2, *J* = 1.8). ^13^C NMR (δ, ppm): 35.81 (C(3)), 56.39 (2 C, 2 3′-MeO), 56.43 (2 C, 2 4′-MeO), 112.10 (2 C(5′)), 113.17 (2 C(2′)), 124.26 (2 C (6′)), 127.06 (2 C(α)H), 128.36 (2 C(1′)), 144,76 (C(2), C(4)), 149.95 (2 C(3′)), 151.76 (2 C(4′)), 190.52 (C(1)). UV–vis (MeCN) *λ*_max_ 401 nm (*ε* = 36,000 M^−1^ cm^−1^), fluorescence (MeCN, *λ*_ex_ 390 nm) *λ*^fl^_max_ 505 nm. IR (CH_2_Cl_2_, ν): 1705 cm^−1^ (C=O). HRMS (ESI+) *m/z* calcd for C_22_H_23_O_5_ [M + H]^+^: 367.1540. Found: 367.1544. Elemental analysis calcd (%) for C_22_H_22_O_5_: C, 72.12, H, 6.05. Found: C, 71.79, H, 6.33. 

**(2*E*,4*E*)-2,4-Bis[4-(methylthio)benzylidene]cyclobutanone (1d)**. A solution of cyclobutanone (35 mg, 0.5 mmol) in EtOH (200 μL) was added dropwise with stirring at 5 °C to a mixture of 4-methylthiobenzaldehyde (157 mg, 1 mmol) and a 0.25 *M* solution of NaOH in an EtOH–H_2_O mixture (2:1 *v*/*v*) (350 μL). The reaction mixture was kept at 5 °C for 20 h, Et_2_O (2 mL) was added, and the resulting thick precipitate was ground to form a powder, which was collected on a filter and recrystallized from EtOH containing some CH_2_Cl_2_. This gave 78 mg of dienone **1d** as a yellow crystalline powder. Yield 46%, m.p. 215–217 °C (from EtOH–CH_2_Cl_2_). ^1^H NMR (δ, ppm, *J*/Hz): 2.51 (s, 6 H, 2 Me,), 3.82 (br.s, 2 H, C(3)H_2_), 7.13 (br.s, 2 H, 2 C(α)H), 7.27 (d, 4 H, 2 H(3′), 2 H(5′), *J* = 8.3), 7.50 (d, 4 H, 2 H(2′), 2 H(6′), *J* = 8.3). ^13^C NMR (δ, ppm): 15.46 (2 C, 2 Me), 36.41 (C(3)), 126.44 (2 C(3′), 2 C(5′)), 126.72 (2 C(α)H), 130.70 (2 C (2′) 2 C (6′)), 131.82 (2 C (4′)), 142.64 (2 C(1′)), 146,04 (C(2), C(4)), 190.59 (C(1)). UV–vis (MeCN) *λ*_max_ 402 nm (*ε* = 39,000 M^−1^ cm^−1^), fluorescence (MeCN, *λ*_ex_ 396 nm) *λ*^fl^_max_ 522 nm. IR (CH_2_Cl_2_, ν): 1708 cm^−1^ (C=O). HRMS (ESI+) *m/z* calcd for C_20_H_19_OS_2_ [M + H]^+^: 339.0871. Found: 339.0869. Elemental analysis calcd (%) for C_20_H_18_OS_2_: C, 70.97, H, 5.36, S, 18.95. Found: C, 70.74, H, 5.50, S, 18.89.

**(2*E*,4*E*)-2,4-Bis[4-(dimethylamino)benzylidene]cyclobutanone (1e).** A 1.33 *M* solution of NaOH in an EtOH–H_2_O mixture (2:1 *v*/*v*) (250 μL) was added with stirring to a mixture of cyclobutanone (35 mg, 0.5 mmol) and 4-dimethylaminobenzaldehyde (149 mg, 1 mmol) in EtOH (250 μL). The reaction mixture was allowed to stand at room temperature for 96 h, and the precipitate was collected on a filter, washed with an EtOH–H_2_O mixture (1:10 *v*/*v*) and Et_2_O, and dried in the air. This gave 120 mg of dienone **1e** as a red-orange crystalline powder. Yield 72%, m.p. 268–271 °C. (cf. Ref. [12]: 274–275 °C). ^1^H NMR (δ, ppm, *J*/Hz): 3.01 (s, 12 H, 4 Me), 3.71 (br. s, 2 H, C(3)H_2_), 6.71 (d, 4 H, 2 H(3′), 2 H(5′), *J* = 8.7), 7.04 (m, 2 H, 2 C(α)H), 7.46 (d, 4 H, 2 H(2′), 2 H(6′), *J* = 8.7). ^13^C NMR (δ, ppm): 35.76 (C(3)), 40.47 (4 C, 4 Me), 112.45 (2 C(3′), 2 C(5′)), 123.31 (2 C (4′)), 126.83 (2 C(α)H), 131.97 (2 C (2′) 2 C (6′)), 142.28 (2 C(1′)), 151,97 (C(2), C(4)), 190.50 (C(1)). UV–vis (MeCN) *λ*_max_ 418 nm (*ε* = 28,000 M^−1^ cm^−1^), 469 nm (*ε* = 46,000 M^−1^ cm^−1^), fluorescence (MeCN, *λ*_ex_ 460 nm) *λ*^fl^_max_ 576 nm. IR (CH_2_Cl_2_, ν): 1695 cm^−1^ (C=O). HRMS (ESI+) *m/z* calcd for C_22_H_25_N_2_O [M + H]^+^: 333.1961 Found: 333.1966. Elemental analysis calcd (%) for C_22_H_24_N_2_O: C, 79.48, H, 7.28, N, 8.43. Found: C, 79.29, H, 7.16, N, 8.20.

**(2*E*,4*E*)-2,4-Bis[4-(diethylamino)benzylidene]cyclobutanone (1f).** A 1.33 *M* solution of NaOH in an EtOH–H_2_O mixture (2:1 *v*/*v*) (250 μL) was added with stirring to a mixture of cyclobutanone (35 mg, 0.5 mmol) and 4-diethylaminobenzaldehyde (177 mg, 1 mmol) in EtOH (250 μL). The reaction mixture was allowed to stand at room temperature for 96 h, and the precipitate was collected on a filter, washed with an EtOH–H_2_O mixture (1:10 *v*/*v*), and dried in air. This gave 145 mg of dienone **1f** as a red-orange crystalline powder. Yield 75%, m.p. 185–187 °C. ^1^H NMR (δ, ppm, *J*/Hz): 1.18 (t, 12 H, 4 Me, *J* = 7.1), 3.40 (q, 8 H, 4 CH_2_N, *J* = 7.1), 3.70 (br.s., 2 H, C(3)H_2_), 6.67 (d, 4 H, 2 H(3′), 2 H(5′), *J* = 8.9), 7.01 (m, 2 H, 2 C(α)H), 7.44 (d, 4 H, 2 H(2′), 2 H(6′), *J* = 8.9). ^13^C NMR (δ, ppm): 12.94 (4 C, 4 Me), 35.70 (C(3)), 45.04 (4 C, 4 CH_2_N), 111.99 (2 C(3′), 2 C(5′)), 122.57 (2 C (4′)), 126.75 (2 C(α)H), 132.28 (2 C (2′) 2 C (6′)), 141.79 (2 C(1′)), 149,53 (C(2), C(4)), 190.44 (C(1)). UV–vis (MeCN) *λ*_max_ 419 nm (*ε* = 33,800 M^−1^ cm^−1^), 481 nm (*ε* = 58,000 M^−1^ cm^−1^), fluorescence (MeCN, *λ*_ex_ 472 nm) *λ*^fl^_max_ 575 nm. IR (CH_2_Cl_2_, ν): 1691 cm^−1^ (C=O). HRMS (ESI+) *m/z* calcd. for C_26_H_33_N_2_O [M + H]^+^: 389.2587. Found: 389.2590. Elemental analysis calcd (%) for C_26_H_32_N_2_O: C, 80.37, H, 8.30, N, 7.21. Found: C, 80.20, H, 8.21, N, 7.07.

### 3.2. Methods

The reactions were monitored by thin layer chromatography using DC–Alufolien Aluminiumoxid 60 F_254_ neutral plates, Merck. The melting points (uncorrected) were determined on a Mel-Temp II instrument. The ^1^H and ^13^C NMR spectra were measured on a Bruker DRX-600 spectrometer (operating at 500.13, 600.22, and 125.76 MHz, respectively) in CD_2_Cl_2_ at 25–30 °C, using the solvent as an internal standard (δ_H_ 5.30 and δ_C_ 54.00 ppm, respectively). The proton and carbon signals were assigned using homonuclear ^1^H-^1^H COSY and heteronuclear ^1^H-^13^C COSY (HSQC and HMBC) 2D spectra. The chemical shifts were determined with an accuracy of 0.01 ppm, and the spin–spin coupling constants were measured with an accuracy of 0.1 Hz.

IR spectra were recorded using the Fourier transform spectrometer Nicolet iS5 (Thermo Fisher Scientific, using an internal reflectance attachment with diamond optical element – attenuated total reflection (ATR, iD7) with 45° angle of incidence. Resolution 4 cm^−1^, the number of scansis 20.

The electronic absorption spectra were recorded on a Cary 4000 spectrophotometer in the MeCN. The fluorescence spectra were obtained on a Cary Eclipse spectrofluorometer at room temperature. All the manipulations with solutions of dyes **1a**–**f** were performed in a darkroom under red light (daylight induces the *E-Z* photoisomerization).

High-resolution mass spectra (HR MS) were run on a Bruker micrOTOF II instrument using electrospray ionization (ESI) [37]. The measurements were done in a positive ion mode (interface capillary voltage of −4500 V), mass range from *m*/*z* 50 to 3000 Da, and external or internal calibration was done with Electrospray Calibrant Solution (Fluka). A syringe injection was used for the solutions in MeCN (flow rate of 3 mL min^−1^). Nitrogen was applied as a dry gas, and the interface temperature was set at 180 °C.

The elemental analysis was carried out at the Microanalytical Laboratory of the A. N. Nesmeyanov Institute of Organoelement Compounds (Russian Academy of Sciences, Moscow, Russian Federation). The samples for the elemental analysis were dried at 80 °C in vacuo.

### 3.3. Cyclic Voltammetry

The electrochemical measurements were carried out using an IPC_Pro M potentiostat in a three-electrode system. A glassy carbon disk (*d* = 2 mm) served as the working electrode, a 0.1 M Bu_4_NClO_4_ solution in MeCN was used as the supporting electrolyte, and an Ag/AgCl/KCl(aq., sat.) reference electrode and a platinum plate auxiliary electrode were used. The working electrode surface was polished by alumina powder with a particle size of less than 0.5 μm (Sigma-Aldrich). In the CV measurements, the potential sweep rate was 100 mV s^−1^. The potentials are presented with *iR*-compensation. The number of transferred electrons was determined by comparing the peak current in the substrate and the current of single-electron oxidation of ferrocene taken in the same concentration. The concentration of compounds **1a**–**d**,**f** was 1 × 10^−4^ M.

### 3.4. X-ray Diffraction Experiments

A suitable single crystal of each of compounds **1b**–**f** and **2** was mounted on a CCD SMART APEX-II diffractometer under a stream of cooled nitrogen, and the crystallographic parameters and X-ray reflection intensities were measured (MoK_α_-radiation (λ = 0.71073 Å), graphite monochromator, ω-scan mode). The reduction of the experimental data was performed using the SAINT program [46].

The structures were solved by direct methods and refined by least squares on *F*^2^ in the anisotropic approximation for non-hydrogen atoms. The hydrogen atom positions were calculated geometrically and refined, at the final stage, using the riding model.

The crystallographic characteristics and structure refinement details are summarized in Appendix A.

The calculations were performed using OLEX-2 and SHELXTL-Plus software [47,48]. The X-ray diffraction studies were done at the Center for Collective Use of the Kurnakov Institute of General and Inorganic Chemistry, Russian Academy of Sciences. The structural data were deposited with the Cambridge Crystallographic Data Centre with numbers: CCDC 2181814 (**1b**), CCDC 2181815 (**1c**), CCDC 2181816 (**1d**), CCDC 1892909 (**1e**), CCDC 2181818 (**1f**), CCDC 2181820 (*syn*,*syn*-**2**) и CCDC 2181819 (*anti*,*anti*-**2**).

### 3.5. Density Functional Theory (DFT) Calculations

The structures and energies of the molecules were calculated using the density-functional theory (DFT) with the PBE0 functional and 6-31 + G(d,p) basis set by the FireFly program [49], partially based on GAMESS code [50]. The solvent (MeCN) effects were taken into account using the dielectric polarizable continuum model (D-PCM) [51]. The vertical absorption and emission spectra and *E*-*E-E*-*Z* isomerization energy profiles were calculated by the time-dependent DFT (TDDFT) with the same functional, basis set, and solvent model. The vertical absorption spectra were calculated by the TDDFT after DFT optimization of the ground state geometry, while the vertical emission spectra were calculated in a similar way after geometry optimization of the π-π* excited state using the TDDFT and D-PCM. The radiative lifetimes were calculated using the formula: *k*_r_ = (⅔)*f*_0i_ν^2^_i0_, τ_r_ = 1/*k*_r_
where *f*_0i_ and ν_i_ are oscillator strength and the frequency of the electronic transition of the *i*th isomer respectively; *k*_r_ is the radiation constant. The isomerization periods were calculated using the formula:*k*_tc_ = *c*ν_i_·exp(−*E*_Ai_/*RT*), t_tc_ = 1/*k*_tc_
where *x_i_* is the vibrational mode frequency of the *i*th isomer, and *E*_Ai_ is the activation barrier of this isomer.

We considered the (*E*,*E*), (*E*,*Z*), and (*Z*,*Z*) isomers of dyes **1a**–**f** and **2**. We have found that the (*E*,*E*) isomers have the lowest energy and account for >99.9% of the isomer mixture. The spectral and ionization properties were calculated only for the (*E*,*E*) isomer.

In dyes **1c** and **2**, free rotation around the C4-C5 bond is possible. The mole fractions of the three possible rotamers of dienones **1c** and **2** were calculated using the partition function:$$x_{i}=e^{-\frac{E_{i}}{kT}}/\sum_{j}e^{-\frac{E_{j}}{kT}},$$
where *x_i_* is the mole fraction of the *i*th conformer and *E_i_* is the ground state energy of this conformer. The calculated UV–Vis absorption and emission properties of the rotamers are almost the same (within 2 nm), therefore, we give only the values for the *syn*,*syn*-conformer.

We simulated the structural relaxation only for the *E*,*E* isomers. It was found that, in the equilibrium mixture of the ground state, the fraction of this isomer is above 90%. It is the main source of photoinduced transformation products.

To construct the profiles of the *E*-*E-E*-*Z* isomerization, we used a simple (unrelaxed) scan of the potential energy surface along the dihedral angle, corresponding to the rotation around the formally double C=C bond with 5-degree increments. The energy values correspond to the non-optimized structures obtained by twisting of the initial isomer.

The left rotation barriers were estimated by the energy difference of the saddle points of the left transition state and stable structures of the *E*,*E* isomers (left minimum). The right ones were estimated as the difference between the top of the right peak of the S1 curve and the local minimum observed on the way to the *E*,*Z* geometry.

We understand that phototransformation, which proceeds via a conical intersection, requires multireference quantum chemistry for an adequate description of the potential energy profiles [52]. Nevertheless, our semi-quantitative description gives insights into the mechanism of phototransformations in organic dyes [53].

The vertical ionization potentials (IP) and electron affinities (EA) were calculated by restricted-open-shell DFT (RO-DFT) for the corresponding monocation and monoanion of each dye. The functional, basis set, and solvation model were the same.

^1^H NMR spectra were calculated using Priroda program package [54,55] with the PBE functional and triple-zeta quality basis set. The optimized geometries were taken from the B3LYP/6-31 + G(d,p)/DPCM calculation. Previously [33], we have shown that, for dienones, the solvent effects are important to properly reproduce the structures and conformation energies.

## 4. Conclusions

The effect of the structure on the photophysical and electrochemical properties of a series of symmetrical dibenzylidene derivatives of cyclobutanone containing electron-donating substituents in the benzene rings of dienones has been studied. It was shown that the products of the condensation of cyclobutanone with benzaldehyde derivatives tend to exist as *E*,*E*-isomers. The conformational analysis of (*E*,*E*)-dienones using X-ray diffraction and NMR data revealed the structural features of these compounds in the crystal and in solution. Quantum chemical calculations confirmed more favorable *syn*,(*syn*/*anti*)-conformations of the conjugated moieties for dibenzylidenecyclobutanones with four methoxy groups. It was found that the PCA reaction of **1a**–**f** in the solid state requires the use of a supramolecular template. Using electronic spectroscopy, the spectral properties of the dienone derivatives were compared. Quantum chemical calculations explained the observed regularities of the luminescence of **1a**–**f** as a function of the donor capacity of the substituent and elucidated the mechanism of luminescence emission and quenching. The dependence of the redox potentials on the position, nature, and number of substituents in the benzene ring, and their correlation with photophysical and quantum chemical characteristics, are of considerable interest for the subsequent investigation of photoactive dienone derivatives. The studied structure and properties of cyclobutanone-based dienones can also be used in the design of photoactive supramolecular systems.

## Data Availability

CCDC No. 2181814 (**1b**), CCDC No. 2181815 (**1c**), CCDC No. 2181816 (**1d**), CCDC No. 1892909 (**1e**), CCDC No. 2181818 (**1f**), CCDC No. 2181820 (*syn*,*syn*-**2**) и CCDC No. 2181819 (*anti*,*anti*-**2**) contain the supplementary crystallographic data for this article. These data can be obtained free of charge at the Cambridge Crystallographic Data Center via www.ccdc.cam.ac.uk/data_request/cif (accessed on 1 October 2022).

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
