# Peer review of "Synthesis, Structure and Photochemistry of Dibenzylidenecyclobutanones"

_molecules, 2022, doi:10.3390/molecules27217602_

Round 1

Reviewer 1 Report

In this work, the author reported synthesis of a series of symmetrical dibenzylidene derivatives of cyclobutanone with E,E geometry. The author also explored the effect of the structure on the photophysical and electrochemical properties the compounds. The compounds are well characterized and even structure of the compounds are proved using single crystal X-ray diffraction. The work is overall very good and potential for publication.

I have two minor comments.

-The author should provide the FT-IR of all the compounds.

-It will be great to know the chemical stability by TGA analysis

-The author should provide the band gap from tac plots calculations.

Thank you

Author Response

Point 1: The author should provide the FT-IR of all the compounds

Response 1: We are grateful to the reviewer for the thorough analysis of the presented material. We agreed with this remark. The FT-IR spectra were recorded for all compounds and made the appropriate corrections in the manuscript. All corrections are highlighted in green.

Point 2: It will be great to know the chemical stability by TGA analysis

Response 2: At the moment we do not have enough material for TGA analysis.

Point 3: The author should provide the band gap from tac plots calculations

Response 3: We cannot answer this remark of the reviewer, because we did not understand what this remark refers to.

Reviewer 2 Report

The work by Fomina and colleagues describes the preparation and photochemical study of a series of dibenzylidenecyclobutanones. Both the experimentation and the computational study are rigorous. In addition, the reported X-ray structures added an extra value to the work.

However, there are are some points that the authors should address in any subsequent revision:

1. 1H and 13C NMR spectra, as well as HRMS spectra of all the compounds should be included in the supporting information. Without these spectra, I can not check the purity of the products. Furthermore, IR spectra of all the compounds should be recorded.

2. I recommend the inclusion of the X-ray tables in the supporting information instead of in the main text of the manuscript.

3. Although the authors explain in the manuscript their insights in the photoisomerization mechanism, I consider that they should include an explanatory scheme of the postulates that they conclude based on their work. This would help the reader to have all the information included in one figure, facilitating the understanding of the authors' conclusions.

4. Regarding the references, there are some messages of errors along the manuscript, probably because of the software that they used:

-Page 2, line 67

-Page 11, table 5 (two errors)

-Page 13, line 345

-Page 17, line 433

5. In the figures 5, 7 and 10, the numbers of the products should be written using bold style.

6. The format of the references must be homogenized.

Author Response

Point 1: 1H and 13C NMR spectra, as well as HRMS spectra of all the compounds should be included in the supporting information. Without these spectra, I can not check the purity of the products. Furthermore, IR spectra of all the compounds should be recorded.

Response 1:  We are grateful to the reviewer for the thorough reviewing of our manuscript and useful comments. We agreed with reviewer's critical notes and made the appropriate corrections in the manuscript. We have included 1H and 13C NMR spectra, as well as HRMS spectra of all the compounds in the supporting information. The FT-IR spectra were recorded for all compounds and made the appropriate corrections in the manuscript.

Point 2:  I recommend the inclusion of the X-ray tables in the supporting information instead of in the main text of the manuscript.

Response 2: We have included X-ray tables in the supporting information.

Point 3: Although the authors explain in the manuscript their insights in the photoisomerization mechanism, I consider that they should include an explanatory scheme of the postulates that they conclude based on their work. This would help the reader to have all the information included in one figure, facilitating the understanding of the authors' conclusions.

Response 3: We have added a scheme of the photoisomerization mechanism to the manuscript.

Point 4: Regarding the references, there are some messages of errors along the manuscript, probably because of the software that they used:

-Page 2, line 67; -Page 11, table 5 (two errors); -Page 13, line 345; -Page 17, line 433

Response 4: We have made the necessary corrections to the manuscript.

Point 5: In the figures 5, 7 and 10, the numbers of the products should be written using bold style.

Response 5: We have made the necessary corrections to the manuscript. All corrections are highlighted in green.

Point 6: The format of the references must be homogenized.

Response 6: We have made the necessary corrections to the manuscript. All corrections are highlighted in green.

Round 2

Reviewer 2 Report

After revision of the new version of the manuscrip, I consider that the the work is suitable for publication in Molecules.

There is only one minor point, I suggest that the authors incorporate the integration of the signals in the 1H NMR spectra included in the supporting information

Author Response

Point 1: I suggest that the authors incorporate the integration of the signals in the 1H NMR spectra included in the supporting information

Response 1: We have incorporated the integration of the signals in the 1H NMR spectra included in the supporting information. All corrections in the manuscript are highlighted in green.